# Developing CRISPR/Cas9-Mediated Fluorescent Reporter Human Pluripotent Stem-Cell Lines for High-Content Screening

**DOI:** 10.3390/molecules27082434

**Published:** 2022-04-09

**Authors:** Kinga Vojnits, Mio Nakanishi, Deanna Porras, Yeonjoon Kim, Zhuohang Feng, Diana Golubeva, Mick Bhatia

**Affiliations:** Department of Biochemistry and Biomedical Sciences, Michael G. DeGroote School of Medicine, McMaster University, Hamilton, ON L8N 3Z5, Canada; vojnitsk@mcmaster.ca (K.V.); nakanim@mcmaster.ca (M.N.); porrasd@mcmaster.ca (D.P.); yekim7211@gmail.com (Y.K.); fengz14@mcmaster.ca (Z.F.); diana.golubeva@stemcell.com (D.G.)

**Keywords:** genome editing, CRISPR/Cas9, OCT4 locus, AAVS1 locus, human pluripotent stem cells, EGFP reporter lines, phenotypic screening

## Abstract

Application of the CRISPR/Cas9 system to knock in fluorescent proteins to endogenous genes of interest in human pluripotent stem cells (hPSCs) has the potential to facilitate hPSC-based disease modeling, drug screening, and optimization of transplantation therapy. To evaluate the capability of fluorescent reporter hPSC lines for high-content screening approaches, we targeted EGFP to the endogenous OCT4 locus. Resulting hPSC–OCT4–EGFP lines generated expressed EGFP coincident with pluripotency markers and could be adapted to multi-well formats for high-content screening (HCS) campaigns. However, after long-term culture, hPSCs transiently lost their EGFP expression. Alternatively, through EGFP knock-in to the AAVS1 locus, we established a stable and consistent EGFP-expressing hPSC–AAVS1–EGFP line that maintained EGFP expression during in vitro hematopoietic and neural differentiation. Thus, hPSC–AAVS1–EGFP-derived sensory neurons could be adapted to a high-content screening platform that can be applied to high-throughput small-molecule screening and drug discovery campaigns. Our observations are consistent with recent findings indicating that high-frequency on-target complexities appear following CRISPR/Cas9 genome editing at the OCT4 locus. In contrast, we demonstrate that the AAVS1 locus is a safe genomic location in hPSCs with high gene expression that does not impact hPSC quality and differentiation. Our findings suggest that the CRISPR/Cas9-integrated AAVS1 system should be applied for generating stable reporter hPSC lines for long-term HCS approaches, and they underscore the importance of careful evaluation and selection of the applied reporter cell lines for HCS purposes.

## 1. Introduction

The derivation of human pluripotent stem cells (hPSCs) [1,2,3,4,5] significantly facilitates stem-cell research and the development of regenerative medicine applications. The versatile application of hPSCs for elucidating regulatory processes during early development [6,7], in vitro disease modeling [8,9,10,11], cell-based therapies [7,12], and drug screening [13,14,15] is based on their robust capability of indefinite self-renewal and broad potential to differentiate into all somatic lineages [16]. 

Chemical genomics approaches such as high-content screening (HCS) and pathway screens of synthetic small molecules and natural products have historically provided useful chemical tool to modulate and study complex cellular processes leading to discoveries of small molecules [17,18,19], e.g., dexamethasone, ascorbic acid, 5-azacytidine, and all-*trans*-retinoic acid, that promote differentiation of various stem cells [20]. Unbiased HCS combines the efficiency of automated high-throughput techniques with the ability of in-depth cellular imaging to collect quantitative data from complex biological systems, providing a method to identify high-quality hits within large compound libraries and facilitate the process of studying biological pathways and finding therapeutic agents for various diseases [21,22]. The multiparametric nature of HCS is particularly well suited for studying complex phenotypes in heterogenous systems, and stem-cell biology has been a focus of recent HCS applications [23,24,25]. HCS can be used all along the preclinical drug discovery pipeline, and it has the power to identify and validate new drug targets or new lead compounds. Most powerfully, HCS can be applied to predict in vivo toxicity, to suggest pathways or molecular targets of lead compounds, and to help predict the efficacy of potential drugs in unique cellular niches when applied to physiologically relevant cellular systems. The most obvious applications of HCS are primary screens of potential leads, i.e., molecules that can be further optimized into drug candidates, for cellular activities that cannot be easily measured by a single endpoint, such as spatially localized proteins or measurements of cellular morphology. For example, HCS can be used to measure cellular differentiation monitored by measuring an increase in the expression of a marker of the differentiated cell type and/or a decrease in the expression of a marker of the undifferentiated cell type. Cellular morphology changes, such as neurite outgrowth, can only be measured in a microscopic image, with or without a molecular marker to confirm the relevance of observed morphology changes.

The predictive power of such systems can often be enhanced by working with primary cells, stem cells, or differentiated stem cells. To harness the full application potential of hPSCs and to further expand their utility in studying gene function and mechanisms in human embryogenesis or human genetic diseases, targeted genome editing with high accuracy and efficiency has long been desired [26,27,28,29,30,31], and extensive and constant efforts have been made in developing novel genome engineering technologies [32,33,34,35]. While classical gene targeting via homologous recombination has previously been inefficient in hPSCs [36,37,38], the development of programmable site-specific nucleases [39,40,41], i.e., engineered “genomic scissors”, such as zinc-finger nucleases (ZFNs) [39,42,43,44,45,46], transcription activator-like effector nucleases (TALENs) [46,47,48,49,50,51,52], and the clustered regularly interspaced short palindromic repeats-associated protein 9 (CRISPR/Cas9) system [53,54,55,56,57,58,59,60,61], has significantly improved hPSC gene targeting by inducing DNA double-strand breaks (DSBs) at desired genomic loci, triggering the endogenous DNA repair machinery [62]. Since CRISPR/Cas9 has been suggested to be easier to engineer, multiplex, and programmable by simply changing the spacer sequence in the synthetic guide RNA (sgRNA), it has emerged as the reference system for genome editing in hPSCs [62,63,64,65,66]. CRISPR/Cas9 has been widely applied in hPSCs for the generation of knockouts [55,67,68,69], knock-ins [70,71], gene silencing/activation [72,73,74,75], genome-wide screening [37,67,76,77,78], chromosome-related studies [79,80,81], disease modeling [82,83,84,85], drug screening [86,87,88], and gene correction therapies [89,90,91,92,93,94], among others.

An exquisite application of gene-editing tools for basic research is the integration of fluorescent protein coding sequences into endogenous genes of interest, generating reporter hPSC lines that facilitate the study of human development in culture and for high-throughput and high-content screening [95,96]. Traditionally, retroviral or lentiviral vectors have been used to generate stable reporter cell lines, but these suffer from frequent transcriptional silencing in hPSC [97]. Moreover, fluorescent proteins, such as GFP which has been most commonly used for studying gene expression, characterization of protein localization, and unraveling cellular signaling pathways via live-cell imaging [98], consist of long sequences, making GFP knock-ins challenging in hPSCs despite the enhancement by CRISPR/Cas9 or other engineered nucleases [71]. Several efforts have been made for efficient insertion of longer sequences, such as utilization of longer homology arms, suppression of key molecules of the nonhomologous end-joining pathway [99,100,101], synchronization with cell-cycle progression [102], using transient antibiotic selection [103], overexpressing RAD51 in the presence of valproic acid [104], or using surrogate reporters [105,106]. However, the efficiency of larger fluorescent protein knock-ins in hPSC is still around 0.83–1.70% [107], which is much lower than the ~20% efficiency observed in somatic cells [107].

Human pluripotent stem cells have the potential to transform the search for new drugs. However, chemical screening campaigns using human stem cells have been limited to diminutive efforts because they are difficult to culture and demonstrate high variability post cell expansion. These complications demand meticulous assay protocols and extreme numbers of replicates that are unprecedented in high-throughput chemical screening. The development of HCS in hPSCs has been challenging due to the difficulties in establishing suitable growth and plating conditions. Here, we report two strategies for the establishment of EGFP-expressing hPSC reporter lines targeting the OCT4 or the AAVS1 locus using the latest CRISPR/Cas9 gene-editing methodologies with specific pros and cons for each approach in hPSC experimentation. Our results provide a foundation for routine applications of HCS assays in hPSC biology and expand the repertoire of fluorescent reporter hPSC lines suitable for HCS and drug discovery. These findings underscore the importance of the careful selection and long-term quality control evaluation of the reporter hPSC lines used in HCS campaigns.

## 2. Results

### 2.1. Generation of OCT4–EGFP Reporter hPSC Line

To establish an OCT4–EGFP reporter system that can be applied as an easy and reliable tool for continuous pluripotency assessment of hPSCs in vitro and in vivo by monitoring endogenous OCT4 expression in living cells, knock-in reporter alleles were generated by targeting the OCT4 locus using drug selection. H9 hPSCs were transduced with three plasmids; one expressed Cas9, while the others targeted OCT4 and contained the fluorescent reporter EGFP (Figure 1A). The designed OCT4-2A-EGFP-PGK-Puro, in which the last OCT4 coding codon is fused in frame with a 2A sequence followed by EGFP and a loxP-flanked puromycin resistance gene expressed from the constitutive PGK promoter, was integrated at the end of the exon 5 of *OCT4* using CRISPR/Cas9 (Figure 1B). EGFP expression and colony formation was detected 2 days after nucleofection, and puromycin selection was applied at day 30. Puromycin selection increased the frequency and intensity of EGFP expression after a few weeks of establishing stable hPSC-like morphology cultures. After colony picking and fluorescence-activated cell sorting (FACS), seven EGFP^+^ clones were isolated. Six clones could be expanded further and analyzed for vectors with correct assembly (Appendix A). Screening of reporter cassette recombination in the OCT4 locus by PCR showed correct integration in clones 5 and 6 (Appendix A), from which clone 5 was selected for further experiments. This selected clone showed strong EGFP expression with typical hPSC morphology co-expressing OCT4 (Figure 1C). This selected hPSC–OCT4–EGFP line continued to express EGFP coincident with the pluripotent marker SSEA-3 (Figure 1D,E, top panels); however, during continuous culture, cells transiently lost their EGFP expression (Figure 1D,E bottom panels), from 98.8% to 0.30%, while the SSEA-3 level remained steady, 35.8% to 37.7%, after 10+ weeks of passaging. Moreover, the endogenous OCT4 level, OCT4 expression intensity, and OCT4^+^ cell number monitored during continuous culture were stable at passages 2 and 10 (Figure 1F). OCT4 expression in the early-passage EGFP^+^ cells was highly comparable to the parental wildtype hPSCs (Appendix A). The stable SSEA-3 and OCT4 levels signify that the hPSC–OCT4–EGFP line could maintain pluripotency but not EGFP expression for long-term culture. It has been previously reported that the presence of a drug-resistance cassette alters proper EGFP expression; however, even after Cre-mediated excision of the PGK-Puro cassette, no EGFP expression was detected in the late-passage (greater than 10 passages) [108] cultures (Figure 1G).

### 2.2. Adapting hPSC–OCT4–EGFP Culture Conditions for High-Content Screening

We next aimed to provide a cost-effective and robust hPSC–OCT4–EGFP-based HCS assay; thus, the sustainable behavior of large-scale early- and late-passage (passages 2 and 8, respectively) cultures in a multi-well format was assessed. Cell count and changes in the expression of OCT4 and EGFP were used as the primary readout (Figure 2A). Undifferentiated colonies were maintained in feeder-free conditions in mTeSR medium and were mechanically dissociated at confluence and plated into 96-well plates at a ratio of one confluent well into one full 96-well plate (Figure 2A). Bone morphogenetic protein 4 (BMP4), as a typical differentiation inducer decreasing OCT4 expression in hPSCs [109], was used to evaluate whether a reduction in both EGFP and OCT4 was quantifiable and could be correlated to cell differentiation. Cells were exposed to BMP4 in a 10 point twofold dilution scheme, starting from 500 ng/mL concentration, for 5 days, while untreated cells as negative controls were maintained in mTeSR medium alone. Images were acquired using an Operetta high-content analyzer; the level of pluripotency marker expression (OCT4–EGFP expression) and cell count (defined by nuclei stained with Hoechst) for each concentration was recorded (Figure 2A). To ensure a significantly large sample size, six fields per well were acquired, which yielded >4000 imaged cells per well. Untreated control cells were grown to confluency per well with typical undifferentiated cell morphology and high OCT4 expression (Figure 2B). From the acquired immunofluorescence images (Figure 2C) through automated image analysis, the generated BMP4 dose–response curves were applied to calculate the half-maximal effective concentration (EC_50_) for each culture, for measured EGFP intensity, OCT4 expression, and cell count (Figure 2D, Table 1). The early-passage culture was significantly more sensitive to BMP4 treatment with lower EC_50_ values (Table 1) than the late-passage culture, and both passages differed from published hPSC BMP4 responses [15]. Furthermore, to evaluate the statistical robustness of the assay, *Z*’ values, a statistical parameter used to compare high-throughput assays [110], were calculated for each screened culture (Figure 2E). A *Z* factor above 0.4 is acceptable, indicating a robust assay. As shown in Figure 2E (top panel), there was a significant difference between the EGFP expression of BMP4-treated versus untreated cells. In contrast, in the later-passage culture, there was too much overlap between the treated (+BMP4) and untreated (−BMP4) wells, resulting in negative *Z*’ values (Figure 2E bottom panel). Overall, these results suggest that our hPSC–OCT4–EGFP reporter line can be adapted to multi-well formats for high-content screening campaigns; however, the loss of EGFP expression during passaging and the altered differentiation behavior of hPSC–OCT4–EGFP represent the main drawbacks of the assay that must be considered. 

### 2.3. Development of a Stable EGFP Reporter hPSC Line Targeting the AAVS1 Locus

Originally described as a major hotspot for adeno-associated virus (AAV) integration, the AAVS1 locus, lying in the first intron of the PPP1R12C gene on human chromosome 19, allows stable and long-term transgene expression in many cell types, including hPSCs. To generate a reporter hPSC line that consistently expresses EGFP in both the undifferentiated state and differentiated derivatives, we targeted to the AAVS1 locus a donor plasmid expressing EGFP under the control of the constitutively active CAG promoter (Figure 3A). After nucleofection and drug selection, colonies were picked and expanded. Four out of the five tested clones had proper insertion as analyzed by PCR (Appendix A). The established EGFP^+^ clones showed strong EGFP expression with typical hPSC morphology co-expressing OCT4 (Figure 3B, Appendix A). The OCT4 expression and nuclear intensity level were comparable to the parental wildtype hPSCs (Appendix A). Moreover, EGFP expression was maintained at a similar level for at least 22 passages or nearly 6 months, without selective pressure (Figure 3C). 

We optimized the hPSC–AAVS1–EGFP line for 96-well plate HCS (Appendix A) and validated it using three defined compounds with established cytotoxicity and stem-cell activity. The levels of pluripotency marker expression (OCT4), EGFP expression, and cell count (defined by nuclei stained with Hoechst) for each compound were recorded using automated microscopy (Appendix A). Comparison of parental wildtype versus the reporter hPSC–AAVS1–EGFP Hoechst^+^, EGFP^+^, and OCT4^+^ cell counts revealed similar EC_50_s and cell behavior during treatment (Appendix A). Moreover, the EGFP^+^ cell count followed the same pattern as Hoechst^+^ cells. These results prove that the hPSC–AAVS1–EGFP line shows similar effects and responses to those seen in the parental wildtype hPSCs, but this reporter line can be superior with faster and cost-effective HCS, as no additional fixing and staining procedures are required.

We next tested whether CAG-driven EGFP expression at the AAVS1 locus could be maintained during differentiation into mesoderm and ectoderm lineages as models for opposite differentiation trajectories. First, we assessed the myelo-erythroid hematopoietic potential of hPSC–AAVS1–EGFP using embryoid body (EB) formation (Figure 3D). By day 10 of differentiation, round and nonadherent hematopoietic cells were observed above the adherent cell layer (Figure 3D) expressing hematopoietic progenitor markers, CD34 and CD45 (Figure 3E). Note, only 37.7% of the CD34^+^/CD45^+^ cells were EGFP^+^ (Figure 3E), which can be related to the heterogenous starting culture also containing non-EGFP^+^ cells. The hPSC–AAVS1–EGFP-derived progenitors presented robust functionality by producing myelo-erythroid colonies (Figure 3F) that still possessed EGFP expression. The hematopoietic differentiation timeline and the derived progenitor morphology and activity were equivalent to published hematopoietic differentiation of normal hPSCs. 

Intermediate stages of neural differentiation were monitored for EGFP expression (Figure 3G); the attached EBs (day 0) and outgrown neural precursors (day 7) retained high EGFP expression levels. In order to set up an HCS platform based on hPSC–AAVS1–EGFP-derived peripheral sensory neurons (SN) that is suitable for drug screening, neural precursors at day 7 were reseeded into 96-well plates and cultured in SN differentiation medium as described in our published protocol [111]. Following differentiation and maturation, SN in 96-well plates showed typical morphology and phenotype for nociceptors expressing the purinergic receptor P2RX3, while maintaining their EGFP expression (days 14 to 66). Thus, the CAG-driven EGFP expression was persistent during in vitro hematopoietic and neural differentiation, indicating that the genomic modification did not impact the pluripotency or differentiation capacity of hPSCs. Overall, the reporter hPSC–AAVS1–EGFP generated here demonstrated faithful robust expression evidenced by persistent EGFP in long-term cell culture and continued to express EGFP in linage-differentiated cells.

## 3. Discussion

The intersection of stem-cell research and genome editing creates expectations and endless promises in revolutionary breakthroughs and fundamental transformation of cell biology, human genetics, and medicine. Since the discovery of hPSCs, the broad application of successful cell replacement therapies and rapid clinical cures has been anticipated; however, now, more than 20 years later, we are still just at the beginning in a journey of understanding the developmental biology and gene function of hPSCs. In parallel, genome-editing technologies have undergone rapid improvement since the CRISPR/Cas9 system was realized in 2013. It is one of the primary topics discussed lately due to its robustness and effectiveness in genome editing, and it has been utilized in laboratories across the world with unlimited possibilities and rash promises. However, with such expectations, pitfalls also emerge, and scientists need to deliver more cautious, quality-controlled results before commitment to specific technological approaches. Solutions are still required to resolve the notorious off-target effects of CRISPR technology, to improve the editing efficiency, and to exploit novel delivery strategies that are safe for clinical stem-cell studies.

Our report evaluates the feasibility of using reporter hPSCs for HCS. We generated two reporter hPSC lines by following two strategies for the establishment of EGFP-expressing hPSC reporter lines targeting the OCT4 or the AAVS1 locus using the latest CRISPR/Cas9 gene-editing methodologies. Both approaches allowed for the efficient generation of reporter lines in approximately 8 weeks. We confirmed that the EGFP reporter is co-expressed with OCT4 with high fluorescent intensity for low-passage cultures. However, in contrast with other studies, we monitored the EGFP expression of hPSC–OCT4–EGFP throughout long-term (more than 10 weeks) passages and realized a significant decrease in EGFP expression. This EGFP loss could have happened due to transcriptional silencing, which has been reported before when using retroviral or lentiviral vectors for hPSC but not with CRISPR/Cas9. The mechanism and reason behind this still need to be evaluated with future studies sequencing the OCT4 locus in the selected clones, and it would be an important and interesting feature of OCT4 knock-in hPSCs. Furthermore, we showed with BMP4 differentiation assays in a high-content screening format that the hPSC–OCT4–EGFP reporter line is more prone to differentiation, indicating that knock-in to the OCT4 locus alters normal hPSC behavior. This behavior was briefly recognized by other groups also mentioning that, e.g., the commercially available H1 OCT4–EGFP reporter hPSC line tends to differentiate more frequently, but this has not been further investigated. Development of high-content screening assays for drug discovery would greatly benefit from a stable EGFP–OCT4 reporter hPSC line, but the continuous EGFP decrease during passaging and the altered differentiation behavior of hPSC–OCT4–EGFP must be resolved. It seems that applying CRIPSR/Cas9 gene editing remains challenging and requires additional solution and evaluation.

In contrast, we successfully generated a hPSC–AAVS1–EGFP reporter line through the combined use of CRISPR/Cas9 and the AAVS1 safe harbor. The AAVS1 safe harbor is one of the very few loci that have been identified to allow transgene expression robustly and stable in nearly all cell types, and it allows robust CAG promoter-driven EGFP expression. Consistent with previous reports, our hPSC–AAVS1–EGFP reporter line expressed EGFP with >50% of the population and still retained ~50% EGFP positivity, even in long-term culture (>22 passages). Moreover, the EGFP expression was maintained during in vitro differentiation from EB formation through lineage maturation; hPSC–AAVS1–EGFP could be differentiated into EGFP-positive hematopoietic progenitors and SN. We demonstrated that both hPSC–AAVS1–EGFP and hPSC–AAVS1–EGFP-derived SN could be adapted to a high-content screening platform that can be applied to high-throughput phenotypic screening campaigns for drug discovery and chemogenomic approaches, i.e., robust biological screens and to elucidate unknown modes of action of neurodevelopmental disorders. 

Our study is the first to compare two EGFP reporter lines generated by CRISPR/Cas9 technology targeting the OCT4 or the AAVS1 loci. Our observations are consistent with recent findings indicating complexity at on-target sites following CRISPR/Cas9 genome editing on the OCT4 loci, such as chromosome instability, on-target mutations, or on-target damage. These could result in phenotypic abnormalities, i.e., continuous EGFP loss and altered differentiation behavior, as shown in this study. In contrast, the AAVS1 locus refers to the region near the first exon and intron of the *PPP1R12C* gene on chromosome 19, which is ubiquitously expressed and considered a safe harbor site. Monoallelic disruption of the *PPP1R12C* gene does not have any adverse effect of the targeted cells, resulting in stable and long-term expression of integrated transgenes in a variety of cell types including hPSCs. For example, as shown by us and other investigators, AAVS1–EGFP expression was persistent and robust in long-term cell cultures. Moreover, after lineage differentiation, differentiated cells still expressed EGFP and were able to maintain high EGFP fluorescence intensity. Thus, the AAVS1 locus serves as a useful site for generation of fluorescent hPSC reporter cell lines that can be applied for long-term HCS approaches beyond 2 weeks.

Human pluripotent stem cells have the potential to transform drug discovery; however, chemical screens using stem cells are limited by throughput or the lack of reliable and stable fluorescent reporter lines. Our results support the use of CRISPR/Cas9 genome-editing technologies to efficiently generate reporter hPSC lines; however, more in-depth long-term studies are needed to assess hPSC behavior during long-term cultures after gene editing to carefully evaluate their feasibility for HCS campaigns.

## 4. Materials and Methods

### 4.1. Pluripotent Stem-Cell Culture and Differentiation

Human pluripotent stem cell (hPSC) research received Canadian Stem Cell Oversight Committee (SCOC; Canadian Institutes of Health Research, CIHR) approval and Research Ethics and Biohazard Utilization Protocols approval at McMaster University, following the principles of the 2016 ISSCR Guidelines for Stem Cell Research and Clinical Applications of Stem Cells. The H9 hPSCs (WAe009-A, WiCell) were maintained in feeder-free culture on Matrigel-coated (BD Biosciences, San Jose CA, USA) plates with mTeSR medium (STEMCELL Technologies, Vancouver BC, Canada) at 37 °C, 5% CO_2_. Cell lines were propagated every 7 days by means of 100 units/mL of Collagenase IV treatment (ThermoFisher Scientific, Waltham MA, USA) for 2–3 min, followed by mechanical dissociation. H9 hPSCs were cultured and differentiated to myelo-erythroid hematopoietic cells and nociceptive sensory neurons as previously described [111,112]. 

### 4.2. Genome Editing

To visualize the expression of endogenous OCT4, CRISPR/Cas9 (Addgene 62205) [113], OCT4-2A-EGFP-PGK-Puro (Addgene 31938) [47], and AAVS1-CAG-EGFP-puromycin [54] donor plasmids were used following published protocols [42,71,114].

### 4.3. Flow Cytometry

H9 hPSC lines in six-well tissue culture plates were treated with 1.5 mL of collagenase and incubated at 37 °C, 5% CO_2_ for 10 min to remove the differentiated cells. Undifferentiated cells were treated with 1.5 mL of Cell Dissociation Buffer (ThermoFisher Scientific) to dissociate into single cells. Then, 4 mL of knockout Dulbecco modified Eagle medium (KO-DMEM) was added to each well. All cells were collected into a single tube and centrifuged at 1500 rpm for 5 min at 4 °C. After the supernatant was aspirated, the pellet was resuspended in 1 mL of PEF medium (PBS with 1 mM EDTA and 3% FBS). Cell Countess was used to count the number of live cells in the cell suspension. A cell density of 1 × 10^5^ was required for every sample to perform flow cytometry. Cells were stained with 7-amino actinomycin (7AAD) (BD Biosciences) to test for cell viability. Live cells were used to analyze cell surface marker expression. SSEA3 (Alexa Fluor 647 Red Anti-SSEA3, BD Biosciences) was used to analyze the pluripotency of the cells, while CD34 and CD45 (BD Biosciences) were used for hematopoietic progenitors. Appropriate negative controls were utilized using fluorescence minus one (FMO) controls. Unconjugated antibodies were visualized with appropriate fluorochrome conjugated secondary antibodies. Flow cytometry was performed on a MACSQuant cytometer (Milteny Biotec, Cologne, Germany) and analyzed using FlowJo software (Tree Star Inc., Ashland OR, USA) 

### 4.4. Immunofluorescence

Immunocytochemical staining was performed with an automated multidrop combi reagent dispenser (ThermoFisher Scientific). Cells were fixed and washed using the BD Cytofix/Cytoperm Fixation/Permeabilization solution kit (ThermoFisher Scientific) containing 4% paraformaldehyde. Cells were incubated with appropriate primary and fluorochrome-conjugated secondary antibodies, and then counterstained with Hoechst 33342 (Invitrogen, Waltham MA, USA). The following antibodies were used: OCT4 (BD Biosciences) and P2X3R (EMD Millipore).

### 4.5. BMP4 Differentiation Assay

A previously published protocol was followed. Briefly, H9 hPSCs cultured in mTeSR were mechanically dissociated at confluence (d7) and plated onto a 96-well black optical plate (Falcon) to a ratio of one confluent well to one full 96-well plate in mTeSR medium. After 24 h, medium was replaced with mTeSR containing BMP4 at 10 point two-fold dilution doses. After 5 days of treatment (6 days in culture), cells were washed with HBSS (ThermoFisher Scientific) and fixed with Cytofix/Permeabilization solution (BD Bioscences). Staining was performed in Cytoperm/Wash solution (BD Biosciences) with the OCT4 Alexa 647 antibody (BD Biosciences, 1:100). Following overnight incubation at 4 °C, cells were washed twice with Cytoperm Wash solution and incubated with 10 μg/mL Hoechst 33342 in Cytoperm wash solution for 10 min at room temperature, followed by three washes with HBSS. 

### 4.6. Screening with hPSC–AAVS1–EGFP Line

Undifferentiated hPSC–AAVS–EGFP and the parental wildtype hPSC lines were mechanically dissociated at confluence and plated onto Matrigel-coated 96-well plates to a ratio of one confluent well to one full 96-well plate in mTeSR medium. Twenty-four hours later, the cells were treated with fresh medium supplemented with the tested compounds, BMP4, Cytarabine, and SCCRI025044, at 10 point two-fold dilution doses starting from 10 μM. Medium with compounds was exchanged daily for 5 days. On day 5, cells were fixed and stained as described above and prepared for automated imaging and plate reader analysis.

### 4.7. Image Analysis

Images were acquired at 10× magnification with an automated high-content confocal fluorescence microscope (Operetta, Perkin Elmer, Woodbridge, ON, Canada) by means of epifluorescence illumination and standard filter sets, and six fields were evaluated for each well. Image analysis was performed using custom scripts in Acapella software (Perkin Elmer). Nuclear objects were segmented from the Hoechst signal. Object intensity analysis was performed on EGFP-positive and OCT4 cells only. Images and well-level data were stored and analyzed in a Columbus Database (Perkin Elmer).

### 4.8. Statistical Analysis

A minimum of three biological replicates was established for each of the described experiments. Statistical analyses were carried out using GraphPad Prism version 7.0a (Graph Pad Software, Inc., Sand Diego CA, USA). All numerical data were expressed as mean values ± SEM or ± SD. Comparisons between two groups were performed using unpaired two-way or one-way Student’s *t*-test assuming two-tailed distribution and unequal variances. For multiple comparisons, ANOVA or Kruskal–Wallis test was applied. Statistical significance was considered at *p* < 0.05, where * *p* = 0.05 and ** *p* = 0.01. 

## Figures and Tables

**Figure 1 molecules-27-02434-f001:**
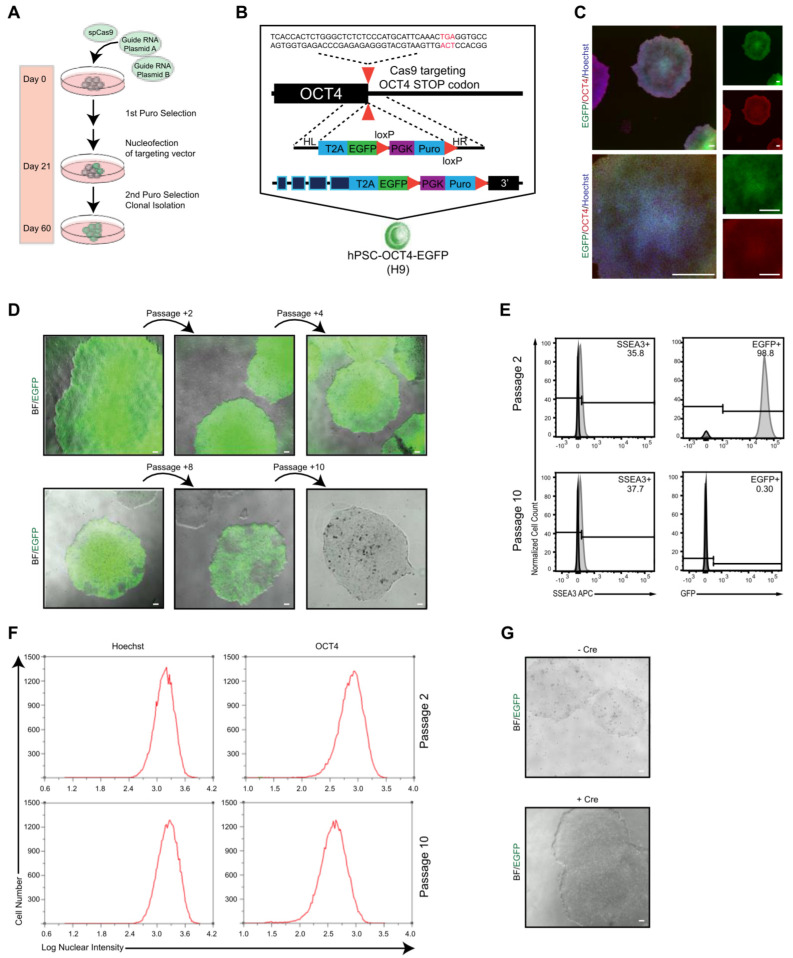
**Generation of OCT4–EGFP hPSC line.** (**A**) Experimental approach for the generation of EGFP knock-in hPSC lines. (**B**) Schematic overview illustrating the targeting strategy for the OCT4 locus. Homologous recombination between the human OCT4 and OCT4-2A-EGFP-LoxP-PGK-Puro-LoxP was promoted by CRISPR/Cas9. Puro, puromycin. (**C**) Immunofluorescence images of undifferentiated hPSC–OCT4–EGFP colonies, cultured in feeder-free condition in mTeSR medium, exhibiting typical cell morphology, and expressing OCT4 (red) and EGFP (green). Nuclei are stained with Hoechst (blue); scale bars = 80 μM and 240 μM, respectively. (**D**) Microscope images showing colony morphology and marker expression of hPSC–OCT4–EGFP cultures after induction and fluorescence-activated cell sorting (FACS) over time with EGFP expression decrease. (**E**) Typical flow histograms show EGFP and SSEA-3 expression levels of hPSC–OCT4–EGFP at early passage (passage 2) and several weeks later (passage 10). (**F**) Histograms show nuclear intensity and frequency of Hoechst^+^ and OCT4^+^ cells at early passage (passage 2) and several weeks later (passage 10). (**G**) Microscope images showing colony morphology and EGFP expression of hPSC–OCT4–EGFP cultures before and after Cre recombinase electroporation.

**Figure 2 molecules-27-02434-f002:**
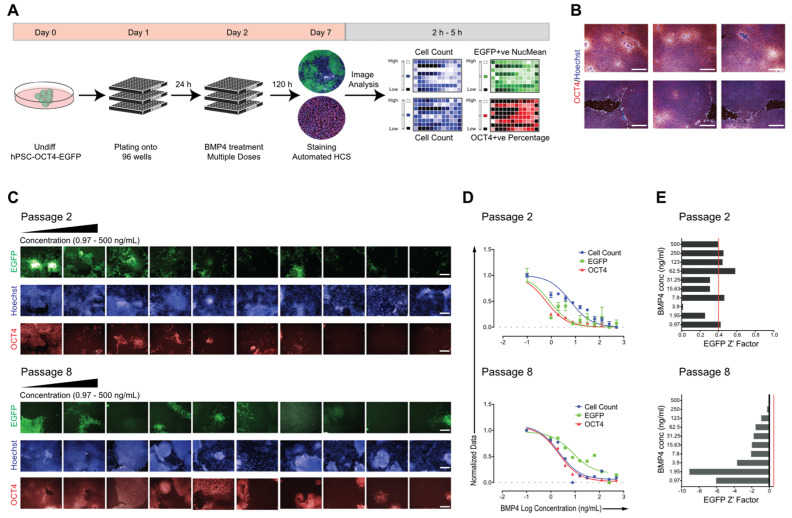
Adaption of hPSC–OCT4–EGFP for high-content screening. (**A**) Schematic representation of the BMP4 dose–response screen on hPSC–OCT4–EGFP. Undifferentiated colonies grown in mTeSR were mechanically dissociated at confluence and plated to 96-well plates at a ratio of one confluent well to one full 96-well plate. After 24 h, BMP4 was added in a 10 point two-fold dilution scheme for an additional 120 h. On day 7, after immunocytochemical staining for OCT4 and Hoechst, automated high-content screening was performed, and the signal intensity and cell count were quantified. (**B**) Representative immunofluorescence images of cells cultured in multi-well formats expressing OCT4 (red). Nuclei are stained with Hoechst (blue); scale bars = 80 μM. (**C**) Representative immunofluorescence images and (**D**) dose–response curves of the responses on early- (passage 2) and late-passage (passage 8) hPSC–OCT4–EGF. Each point: *n* = 3, mean ± SEM, scale bars = 80 μM. (**E**) For each tested BMP4 concentration, the *Z* factor was calculated. The indicated *Z* factor acceptance threshold is 0.4.

**Figure 3 molecules-27-02434-f003:**
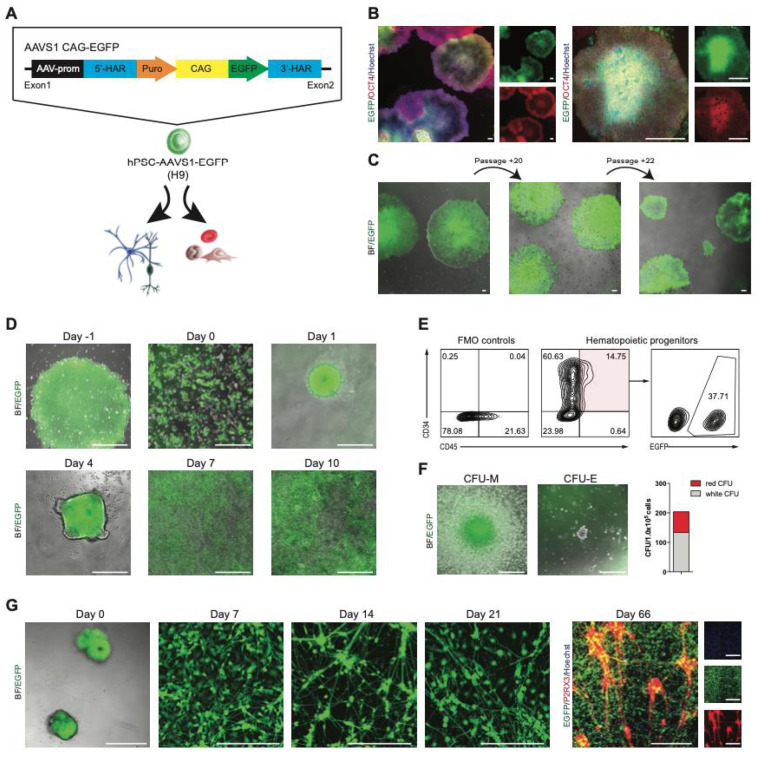
Hematopoietic and neural differentiation of hPSC–AAVS1–EGFP. (**A**) Experimental approach for the generation of EGFP knock-in hPSC lines, and illustration of the AAVS1 EGFP reporter allele. 5′-HAR/3′-HAR, upstream/downstream homology arm; puro, puromycin; CAG, CAG promoter. (**B**) Immunofluorescence images of undifferentiated hPSC–AAVS1–EGFP colonies, cultured in feeder-free conditions in mTeSR medium, exhibiting typical cell morphology, and expressing OCT4 (red) and EGFP (green). Nuclei are stained with Hoechst (blue); scale bars = 80 μM and 240 μM, respectively. (**C**) Microscope images showing colony morphology and marker expression of hPSC–AAVS1–EGFP cultures after induction and FACS over time with stable EGFP expression. (**D**) Phase-contrast and immunofluorescence images of differentiating hPSC–AAVS1–EGFP cells; scale bars = 240 μM. (**E**) Flow cytometry of total (CD34^+^/CD45^+^) and EGFP^+^ hematopoietic progenitors. (**F**) Microscopic images show different colony types in methylcellulose assay; scale bars = 200 μM. Bar graph indicates colony-forming activity; *n* = 1. (**G**) Phase-contrast and immunofluorescence images of sensory neural (PNS) differentiation of hPSC–AAVS1–EGFP from EB formation (day 0) through sensory neural maturation until day 66. Cultures were stained positive for EGFP (green) and for PNS-specific purinergic receptor (P2RX3, red). Nuclei were stained with Hoechst (blue); scale bars = 240 μM.

**Table 1 molecules-27-02434-t001:** Comparison of the calculated half-maximal effective concentration (EC_50_) of BMP4 tested on early- (passage 2) and late-passage (passage 8) hPSC–OCT4–EGFP cultures.

	Passage 2 EC_50_ (ng/mL)	Passage 8 EC_50_ (ng/mL)
Cell count	5.6	3.75
EGFP	1.23	7.8
OCT4	0.78	3.52

## Data Availability

Data presented in this study available in the main or Appendix A and are available on request from the corresponding author.

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
