# Peer review of "Developing CRISPR/Cas9-Mediated Fluorescent Reporter Human Pluripotent Stem-Cell Lines for High-Content Screening"

_molecules, 2022, doi:10.3390/molecules27082434_

Round 1

Reviewer 1 Report

In the presented work authors described an establishment and characterization of 2 reporter cell lines derived from human embryonic stem cells.

All the results are clearly presented and adequately discussed, methods are described in sufficient details. I do not have any major objections regarding this study and I think the paper is eligible to be published in the Molecules journal.

I have some minor points/comments.

1) Results for EGFP reporter inserted into the Oct4 locus were done on one selected clone. Authors described differences in differentiation potential compared to parental (wild-type) ES cells. Such a difference (also described in literature for other ES cell clones, even commercially available) may be a result of EGFP reporter itself or of a haploinsufficiency of the Oct-4 gene (as inserted reporter disrupts one allele). Did authors checked for alterations of differentiation potential also in clones where Oct-4 locus was just disrupted (without EGFP reporter expression) thus mimicking “pure” haploinsufficiency state? Are the observed results supported by an analysis of different clone(s) (or really just the selected one was analyzed)? The haplonisufficency problem is also supported by results in Fig. 1F, where Oct-4 expression is decreased during passaging the cells – did authors compared Oct-4 expression also to parental ES cells (whether it is decreased even at early passage 2)?

2) Observed functional differences might be caused by non-specific effects of Cas9 (as is properly discussed in the paper). Did authors analyzed/sequenced Oct-4 locus in the clone selected for experiments?

3) Both reporter cell lines described in the paper, Oct4-EGFP and AAVS1-EGFP, have different behavior and potential utilization as Oct4-reporter could be used to monitor pluripotency/differentiation potential, but AAVS1-reporter is expressed constitutively from the CAG promoter. Did authors try to insert EGFP reporter under the control of (endogenous) Oct-4 promoter into the AAVS1 locus to mimic the situation/function of the OCT4-EGFP reporter?

4) A typo is present in line 333: …through the combined …

5) Although not completely eligible to check English grammar, lines 392-394 might be reviewed.

Reviewer 2 Report

The Authors present data about the use of CRISPR/Cas9 system to knock-in fluorescent protein to endogenous genes of interest in human pluripotent stem cells (hPSC). They targeted EGFP to the endogenous OCT4 locus and to the AAVS1 locus on hPSC lines. They also tested the EGFP expressing hPSC-OCT4-EGFP line in high-content screening assays, but they reported the loss of EGFP expression during passaging and the altered differentiation behavior of cells, this impairing their use in such assays. They suggest the CRISPR/Cas9 integrated AAVS1 system as a valuable approach to generate reporter cell lines for HCS purposes, due to the more stable and consistent EGFP expression, with no negative influence on PSCs expansion and differentiation.

Major points:

  1. The source of H9 hPSC cells should be described in the materials and methods
  2. The correct insertion of donor DNA at the OCT4 or AAVS1 locus should be verified by PCR of gDNA extracted from individual clones and Sanger sequencing using specifically designed primers
  3. To strengthen the authors’ conclusions, it would be useful to perform adaption for high-content screening also for hPSC-AAVS1-EGFP cell line.

Round 2

Reviewer 2 Report

The Authors satisfactorily addressed my questions, by adding supplemental figures and by improving the manuscript text.